# ASK QUESTION WITH DOUBLE HINTS: VISUAL QUESTION GENERATION WITH ANSWER-AWARENESS AND REGION-REFERENCE

## ABSTRACT

The task of visual question generation (VQG) aims to generate human-like questions from an image and potentially other side information (e.g. answer type or the answer itself). Despite promising results have been achieved, previous works on VQG either i) suffer from one image to many questions mapping problem rendering the failure of generating referential and meaningful questions from an image, or ii) ignore rich correlations among the visual objects in an image and potential interactions between the side information and image. To address these limitations, we first propose a novel learning paradigm to generate visual questions with answer-awareness and region-reference. In particular, we aim to ask the right visual questions with *Double Hints - textual answers and visual regions of interests*, effectively mitigating the existing one-to-many mapping issue. To this end, we develop a simple methodology to self-learn the visual hints without introducing any additional human annotations. Furthermore, to capture these sophisticated relationships, we propose a new double-hints guided Graph-to-Sequence learning framework that first models them as a dynamic graph and learns the implicit topology end-to-end, and then utilize a graph-to-sequence model to generate the questions with double hints. Our experiments on VQA2.0 and COCO-QA datasets demonstrate that our proposed model on this new setting can significantly outperform existing state-of-the-art baselines by a large margin.

## 1 INTRODUCTION

Visual Question Generation (VQG) is an emerging task in both computer vision (CV) and natural language processing (NLP) fields, which aims to generate human-like questions from an image and potentially other side information (e.g. answer type or answer itself). Recent years have seen a surge of interests in VQG because it is particularly useful for providing high-quality synthetic training data for visual question answering (VQA) (Li et al., 2018) and visual dialog system (Jain et al., 2018). Conceptually, it is a challenging task because the generated questions are not only required to be consistent with the image content but also meaningful and answerable to human.

Despite promising results have been achieved, previous works still encounter two major issues. First, all of existing methods significantly suffer from one image to many questions mapping problem rendering the failure of generating referential and meaningful questions from an image. The existing VQG methods can be generally categorized into three classes with respect to what hints are used for generating visual questions: 1) the whole image as the only context input (Mora et al., 2016); 2) the whole image and the desired answers (Li et al., 2018); 3) the whole image with the desired answer types (Krishna et al., 2019). Since a picture is worth a thousand words, it can be potentially mapped to many different questions, leading to the generation of diverse non-informative questions with poor quality. Even with the answer type or desired answer information, the similar one-to-many mapping issue remains, partially because the answer hints are often very short or too broad. As a result, these side information are often not informative enough for guiding question generation process, rendering the failure of generating referential and meaningful questions from an image.

The second severe issue for the existing VQG methods is that they ignore the rich correlations among the visual objects in an image and potential interactions between the side information and

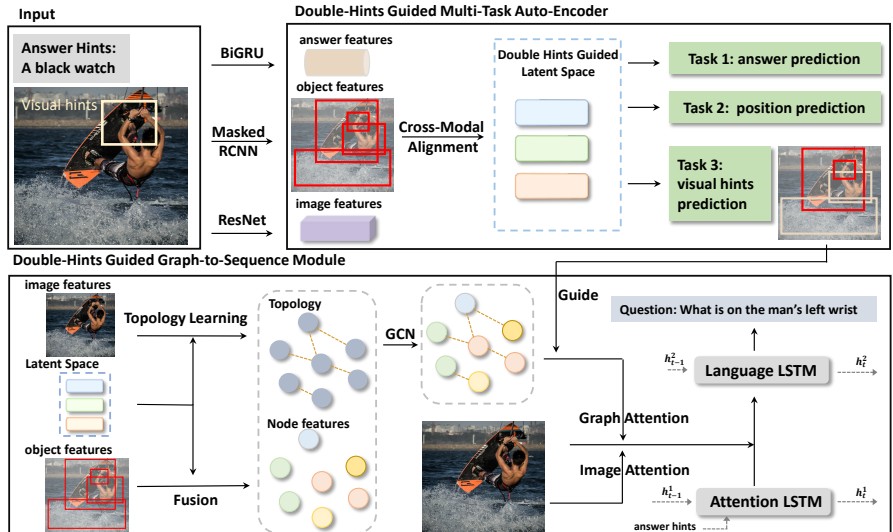

Figure 1: The overall framework of our proposed model with double hints to guide VQG.

image (Krishna et al., 2019). Conceptually, the implicit relations among the visual objects (e.g., spatial, semantic) could be the key to generate meaningful and high-quality questions. This is partially because when human annotators ask questions about a given image, they often focus on these kinds of interactions. In addition, another important factor for producing informative and referential questions is about how to make full use of side information to align with the targeted image. Modeling such potential interactions between the side information and an image becomes a critical component for generating referential and meaningful questions.

To address these aforementioned issues, in this paper, we first propose a novel learning paradigm to generate visual questions with answer-awareness and region-reference. More specifically, we aim to utilize the referential visual regions of interest hints (denoted as visual hints for simplicity) of the images and the textual answers (denoted as answer hints) to faithfully guide question generation. As illustrated in Figure 1, by giving an image with visual hints (the region enclosed by the orange rectangle) and answer hints (the answer), the model is able to faithfully generate the right question with key entities that reflects the visual hints and answerable to the answer hints. To this end, in order to learn these visual hints, we develop a multi-task auto-encoder to learn the visual hints and the unique attributes automatically without introducing any additional human annotations.

Furthermore, to capture the rich interactions between visual and answer hints and the image as well as the sophisticated relationships among the visual objects in an image, we propose a new Double-Hints guided Graph-to-Sequence learning framework (DH-Graph2Seq). The proposed model first models these interactions as a dynamic graph and learns the implicit topology end-to-end, and then utilize a Graph2Seq model to generate the questions with double hints. In addition, in the decoder side, we also present a visual-hint guided separate attention mechanism to attend image and object graph separately and overlook the non-visual-hints particularly.

In summary, we highlight our main contributions as follows:

- We propose a novel learning paradigm to generate visual questions with *Double Hints - textual answer and visual regions of interests*, which could effectively mitigate the existing one-to-many mapping issue. To the best of our knowledge, this is the first time both visual hints and answers hints are used for the VQG task.

- We explicitly cast the VQG task as a Graph-to-Sequence (Graph2Seq) learning problem. We employ graph learning technique to learn the implicit graph topology to capture various rich interactions between and within an image, and then utilize a Graph2Seq model to guide question generation with double hints.

- Our extensive experiments on VQA2.0 and COCO-QA datasets demonstrate that our proposed model can significantly outperform existing state-of-the-art by a large margin.

## 2 RELATED WORK

### 2.1 VISUAL QUESTION GENERATION

Visual question generation is an emerging task in the visual-language domain. Mora et al. (2016) firstly makes an attempt to adapt the CNN-LSTM model to generate image-related question and answer pairs. These works simply map the visual features into textural questions, which lead to abstract and general results (Mostafazadeh et al., 2016; Zhang et al., 2016). Jain et al. (Jain et al., 2017) use variational autoencoder together with an LSTM decoder to generate diverse questions. What's more, dual learning is applied in VQG task (Li et al., 2018; Shah et al., 2019). They view VQG as the dual task of VQA, thus optimizing both of them according to cycle-consistency. Liu et al. (2018) view this task the inverse of VQA and combining the type-specific partial question with the question-answer pairs to produce more reasonable results. Furthermore, Krishna et al. (Krishna et al., 2019) introduce the fine-grained answer type as the guidance to the variational method, which generates goal-driven questions. Xu et al. (2020) adapt the graph method to generate meaningful questions with the target of answers. Different from these works, we will generate the questions under the guidance of both visual regions of interest and textual answers, which can generate questions more referential and answerable.

### 2.2 GRAPH-TO-SEQUENCE LEARNING

Graph neural networks (GNN) (Kipf & Welling, 2016; Gilmer et al., 2017) has drawn a significant amount of attention in recent years. In the NLP domain, graph-to-sequence learning is to generate sequential results from graph-structured data, which mapping the structural data to sequence output (Xu et al., 2018; Chen et al., 2019c; Gao et al., 2019). When coming to the non-graph structured data just like the regions of an image, researchers explore some methods to construct objects' (Norcliffe-Brown et al., 2018) or words' (Chen et al., 2019b;c) topology.

Unlike these previous methods, we propose a novel cross-modal graph2seq model which models the relations upon the visual and textural hints. With the two modal's guidance, the model will learn the appropriate embedding which is crucial to generate high-quality questions.

## 3 METHOD

We first introduce our novel learning paradigm that generates visual questions with double hints. And then we present the details of each component of our proposed model as shown in Figure 1.

### 3.1 PROBLEM FORMULATION

The goal of the visual question generation task is to generate human-like and answerable natural language questions based on the given images and potentially other side information, such as textural answer and answer type, where the generated questions need to be consistent with the given images and the hints semantically. In this paper, we will first discuss the traditional answer-based hints and then present our new learning paradigm.

We assume that the raw image is $I$, and the target answer (answer hint) is a collection of word tokens represented by $X^a = \{a_1, a_2, ..., a_m\}$, where $m$ denotes the length of the answer. The task of traditional answer-guided VQG is to generate the natural language question consisting of a sequence of word tokens $\hat{X}^q = \{q_1, q_2, ..., q_n\}$ which maximizes the conditional likelihood $\hat{X}^q = argmax_{X^q} P(X^q|I, X^a)$, where $n$ is the number of word tokens. To address the proposed issues, we introduce a new setting that focuses not only on the answer-awareness but also on region-reference. Specifically, we formulate it as a joint optimization over visual hints finding and double-hints guided graph2seq learning tasks. The visual hint is a collection of visual object regions of interest in the image which are direct visual clues for question generation. We denote it as $V = \{v_1, v_2, ..., v_N\}$, where $v_i$ is the bounding box in the image and $N$ is the number of visual hints. Under this setting, the likelihood can be cast as $P(X^q|I, X^a) = P(X^q|V, I, X^a)P(V|I, X^a)$. In particular, the visual hints will be generated in data pre-processing (see Sec. 4.1) and the model can learn them without human annotations. Thus there are no visual hints during inference.

To make the statement clear, we will give mathematical notations. We denote the input image feature as $\mathbf{I} \in \mathbb{R}^{c \times F_v}$, where $c$ is the channel number and $F_v$ is the visual feature dimension. Each image is given a collection of object regions denoted by $\mathbf{V} = \{v_1, v_2, .., v_N\}$, where $v_i$ is associated with its visual feature $\mathbf{r}_i \in \mathbb{R}^{F_v}(\mathbf{r}_i \in \mathbf{R})$, bounding box position vector $\mathbf{p}_i = [x_0, y_0, x_1, y_1]$, category attributes (eg. it belongs to animal) $\mathbf{c}_i$ and visual hint indicator $\hat{\mathbf{y}}_i$ ($\hat{\mathbf{y}}_i$ is true iff $v_i$ is the visual hint during training stage). $(x_0, y_0), (x_1, y_1)$ are the normalized up-left and bottom-right coordinates. As for the textural answers, we adopt a GRU module to encode the answer embedding $X^a$ with $m$ words to learn the word semantic features $\mathbf{X}^a = \{\mathbf{a}_1, \mathbf{a}_2, ..., \mathbf{a}_m\} \in \mathbb{R}^{m \times F_a}$. $F_a$ denotes the answer vector dimension.

## 3.2 Double-Hints Guided Multi-Task Auto-Encoder

The most important point of our first issue located in how to effectively find the visual hints and how to combine the visual features with double hints. In this section, we firstly introduce a cross-modal alignment that aligns the cross-modal visual objects and textural answers into the latent space. Then we propose a multi-task decoder that can learn the visual hints as well as keep the crucial attributes in the latent space.

### 3.2.1 Cross-modal Alignments

To infer which regions of interest are suitable for asking questions, answer clues and visual object features are both crucial. To exploit the fine-grained interactions between visual clues and textural answer hints, we explicitly model the global correlations between them in the embedding space.

Firstly, we notice that the position and category attributes of objects are indispensable during fine-grained object relation modeling. So we incorporate them by projecting the position vector $\mathbf{p}_i$ and category attribute $\mathbf{c}_i$ for object $v_i$ into two embedding space (we assume their dimensions are $d$). Then we concatenate them with visual feature $\mathbf{r}_i$. We overload $\mathbf{r}_i$ here considering the conciseness, which means in the following content, $\mathbf{r}_i \in \mathbb{R}^{F_V}$ ($\mathbf{r}_i \in \mathbf{R}, F_V = F_v + 2*d$) denotes the object features with position and category embedding.

For each object $\mathbf{r}_i$ and answer word $\mathbf{a}_j$, we will calculate the alignment score $\mathbf{S}_{ij}$ as follows:

$$\mathbf{S}_{ij} = \sigma(\mathbf{r}_i \mathbf{W}_r + \mathbf{a}_j \mathbf{W}_a)\mathbf{W} \tag{1}$$

where $\mathbf{W}_r \in \mathbb{R}^{F_V \times F_{align}}$, $\mathbf{W}_a \in \mathbb{R}^{F_a \times F_{align}}$ and $\mathbf{W} \in \mathbb{R}^{F_{align}}$, $F_{align}$ is the hidden dimension size. What's more, $\sigma$ denotes the Tanh function. So the alignment score matrix $\mathbf{S} \in R^{N \times m}$ represents the correlations between the objects and the answer words. Then we align the objects with answer according to $\mathbf{S}$.

$$\mathbf{H} = \text{Align}(\mathbf{R}, \mathbf{X}^a) = \text{CONCAT}(\mathbf{R}; \mathbf{X}_{align}), \mathbf{X}_{align} = \beta \mathbf{X}^a, \beta_{ij} = \frac{exp(\mathbf{S}_{ij})}{\sum_{j=0}^{j=m} exp(\mathbf{S}_{ij})} \tag{2}$$

where $\mathbf{H}$ is the aligned objects' representation and CONCAT is the row-wise concatenation operation.

### 3.2.2 Multi-Task Decoder

After aligning the visual features with the answers, we introduce the multi-task decoder which can infer the visual hints and the unique attributes (i.e object position and answers). By this, the latent embedding absorbs the double hints while retaining the robust features like objects' position clues.

Firstly, we apply MLP with ReLU activation to project $\mathbf{H}$ to a $F_h$ dimension latent space denoted as $\mathbf{L}$. Empirically, it is not proper to infer the attributes directly by $\mathbf{L}_i$ individually. To infer the visual hints better, we capture the top-down signals (Anderson et al., 2018) by attending to the image features. In other words, it is more reasonable to combine the fine-grained object features with the global coarse-grained image features. So we apply the attention mechanism between object $\mathbf{L}_i$ and the whole image $\mathbf{I}$.

$$\mathbf{L}_{attn} = \beta \mathbf{I} \mathbf{W}_{attn}, \beta_{i,j} = \frac{exp(\mathbf{S}_{ij})}{\sum_{j=0}^{j=m} exp(\mathbf{S}_{ij})}, \mathbf{S}_{ij} = \sigma(\mathbf{L}_i \mathbf{W}_l + \mathbf{I}_j \mathbf{W}_I)\mathbf{W} \tag{3}$$

where $\mathbf{W}_l \in \mathbb{R}^{F_h \times F_{hid}}$, $\mathbf{W}_I \in \mathbb{R}^{F_v \times F_{hid}}$, $\mathbf{W} \in \mathbb{R}^{F_{hid}}$, $\mathbf{W}_{attn} \in \mathbb{R}^{F_v \times F_h}$, $\boldsymbol{\beta} \in \mathbb{R}^{N \times c}$ and $\mathbf{I} \in \mathbb{R}^{c \times F_v}$. Then we use residual connection to combine latent features and global features as $\mathbf{L}_{enh} = (\mathbf{L} + \mathbf{L}_{attn})/\sqrt{2}$. In the following paragraph, we will discuss the the multi-task decoder on the latent space $\mathbf{L}_{enh}$ to generate the visual hints and keep the unique attributes in the latent embedding.

**Visual hint prediction**    We apply a feed-forward layer and the softmax function to predict the probability of visual hints denoted as $\mathbf{p}_{vh}$. In practice, the number of visual-hint objects are much smaller than non-visual-hint objects, so we combine balanced-cross-entropy loss and focal loss (Lin et al., 2017) together as follows:

$$L_{vh} = -\frac{\eta}{N_{pos}}\sum_i \hat{\mathbf{y}}_i \mathbf{p}_{vh,i}^\lambda log(\mathbf{p}_{vh,i}) - \frac{\eta}{N_{neg}}\sum_i (1-\hat{\mathbf{y}}_i)(1-\mathbf{p}_{vh,i})^\lambda log(1-\mathbf{p}_{vh,i}) \quad (4)$$

where $N_{neg}$ denotes the number of non-visual-hints and $N_{pos}$ is the number of visual hints.

**Object position prediction**    For high-quality question generation, the relative spatial relations among objects are important clues. So we want to keep these essential features in the latent space via predicting the coordinates of the objects. Just like the previous visual hints prediction, we apply the feed-forward layer on $\mathbf{L}_{enh}$ and predict the absolute normalized position vector $\mathbf{p}_{pos}$. The loss $L_{pos}$ will be the mean square loss function of $\mathbf{p}_{pos}$ and ground-truth $\mathbf{p}$.

**Target answer prediction**    As the answer hint is the guidance of asking questions, we ensure that the latent embedding indeed retains it via regenerating the answer. Technically, we apply a dilated CNN with a max-pooling layer to get the global representation of the latent space, then we predict the answer we expect to ask. We denote the answer is $t_i \in \{t_0, t_1, ..., t_c\}$, where $i$ is the $i-th$ sample, and $c$ is the amount of all answers in the training dataset.

$$\mathbf{L}_{pool} = \text{MaxPool}(\text{ReLU}(\text{DilatedCNN}(\mathbf{L}_{enh}))) \quad (5)$$

Then we apply the feed-forward layer on $\mathbf{L}_{pool}$ and softmax function to calculate the probability. Finally, we adopt the cross-entropy loss to calculate the answer prediction loss denoted as $L_{ans}$.

## 3.3 DOUBLE-HINTS GUIDED GRAPH CONSTRUCTION AND EMBEDDING

To capture the complex correlations among visual objects in the image, we regard the objects as nodes in the object graph $\mathcal{G}$ and adopt the paradigm of graph learning. Most common GNN applications often employ a GNN algorithm on structured data to capture the rich structure information and compute the graph node embedding directly. But for VQG, the objects are unstructured, whose relations are not explicitly given. So in this section, we construct the object graph $\mathcal{G}$'s topology with the guidance of double hints and then employ a GNN model to encode them.

The topology of a graph represents the relations of the nodes. We claim that under our setting, the graph edges should be weighted and learnable (i.e $\mathcal{G}$ is a dynamic graph) because the relation of two objects should be decided by the double hints. We take the latent embedding $\mathbf{L}_{enh}$ of multi-task auto-encoder to exploit such topology since the objects' visual features incorporate the double hints well by the auto-encoder.

Inspired by (Chen et al., 2019a), a good similarity metric function is supposed to be learnable and expressively powerful. We first calculate the dense similarity matrix $\mathbf{S}$ by multi-head weighted cosine similarity function as follows:

$$s_{ij}^p = cos(\mathbf{w}_p \odot \mathbf{L}_{enh,i}, \mathbf{w}_p \odot \mathbf{L}_{enh,j}), \mathbf{S}_{i,j} = \frac{\sum_{p=1}^k s_{i,j}^p}{k} \quad (6)$$

where $\mathbf{w}_p$ is the learnable weight, $\odot$ is the Hadamard product, and $s_{i,j}^p$ denotes the similarity between node $i$ and $j$ of head $p$. The model learns to highlight specific dimensions of the latent embedding space by such formula. $k$ is the heads' number. The final similarity results will be the mean of the heads. Because the learned graph similarity matrix $\mathbf{S}$ is dense and ranges between [-1, 1], so we adopt $\epsilon-$sparsing to make it non-negative and sparse. Specifically, we mask off $\mathbf{S}_{i,j}$ (i.e., set to zero) if it is smaller than the threshold $\epsilon$ to get the adjacency matrix $\mathbf{A}$.

Then we apply a multi-layer graph convolution network (GCN) to effectively learn the node embedding from the constructed object graph. Firstly, we align the object feature $\mathbf{R}$ with the learned double hints. Similarly, the multi-task auto-encoder has fused the double hints and predicted the visual hints. Thus it's latent embedding is the exact guidance we want. The node features will be represented as follows:

$$\mathbf{X} = \mathrm{F}(\mathbf{R}; \mathbf{X}^a) = \mathrm{CONCAT}(\mathbf{R}; \mathrm{Latent}(\mathbf{R}; \mathbf{X}^a)) = \mathrm{CONCAT}(\mathbf{R}; \mathbf{L}_{enh}) \qquad (7)$$

where Latent is the auto-encoder's latent space representation in Sec. 3.2. And CONCAT is the node-wise concatenation. Then we stack $k$ layers of classic spectral GCN (Kipf & Welling, 2016) together with residual architecture to aggregate the fine-grained region features. Appendix A is referred to the readers for more details.

### 3.4 DOUBLE-HINTS GUIDED QUESTION GENERATION MODULE

For question generation, we adapt the language model from (Lu et al., 2018) for the image, graph with double hints. The model consists of two LSTMs whose initial states are initialized by answer features: the first one for encoding global image feature and the word embedding $\mathbf{q}_t$ into the hidden state $\mathbf{h}_1^t \in \mathbb{R}^m$ where $m$ is the dimension and the second one for question words prediction. Between the two LSTMs, the model will attend the image and graph separately by the visual hints. We refer to this procedure as visual-hint-guided separate attention.

The visual-hint-guided separate attention takes in the image and graph and attended by hidden state $\mathbf{h}_1^t$ separately. Specifically for graph attention, we ignore the nodes which are predicted to be non-visual-hint. We name this mechanism as visual attention as follows:

$$\mathbf{X}_{vh} = \mathrm{VisualHintMask}(\mathbf{X})$$
$$\mathbf{h}_{graph} = \mathrm{Attention}(\mathbf{X}_{vh}, \mathbf{h}_1^t) \qquad (8)$$

where *VisualHintMask* is to mask off the non-visual-hint objects, $\mathbf{X}$ is the graph node embedding and *Attention* is the classic attention mechanism (Bahdanau et al., 2014). We train with the cross-entropy loss $L_{ques}$.

The final loss will be the combination of the previous components.

$$L = L_{ques} + \alpha L_{vh} + \beta L_{pos} + \gamma L_{ans} \qquad (9)$$

## 4 EXPERIMENTS

In this section, we evaluate the effectiveness of our proposed model. The code and data for our model are provided for research purposes[1].

### 4.1 DATSETS AND PRE-PROCESSING

**Datasets** We conduct the experiments on the VQA2.0 (Antol et al., 2015) and COCO-QA (Ren et al., 2015) datasets. Both of them use images from MS-COCO (Lin et al., 2014). And to fit our setting, we introduce a simple method to generate the visual hints of the original pairs (image, question, answer) without human annotations: (1) we use the Masked-RCNN to generate objects (category attributes) $V$ in the image. (2) we use standfordcoreNLP to find the noun-words in both questions and answers. For each object attributes and noun-words we use the GloVe model to initialize them and take an average to get the vector representation denoted by $\mathbf{g}_{obj}$ and $\mathbf{g}_{noun}$. The visual-hint indicator $\hat{\mathbf{y}}_i$ is positive iff its' l2 distance with any $\mathbf{g}_{noun}$ is larger than the threshold $\mu$. But there are two special cases that can lead to no aligned objects: (1) there are exactly no visual hints (eg. Q: Is there any book? A: No) (2) the error caused by the detection model or the NLP tools leads to no visual hints. For the first case, we will drop them due to the technical drawback. For the second case, we will keep them. Moreover, although one image could have multiple answer hints (one image-answer pair generally corresponds to one question), there are a small portion of image-answer pair linking to multiple questions. In this case, we will randomly reserve one. After processing, the VQA 2.0 contains 239973 examples for training split and 116942 for validation. And COCO-QA contains 53440 for training and 26405 for validation.

---

[1]Code will be released upon the paper acceptance.

Table 1: Results on VQA2.0 and COCO-QA val set. All accuracies are in %. Notations: B@4-BLEU@4, C-CIDEr, M-METEOR, R-ROUGE, QB@1-Q-BLEU@1.

| Dataset | VQA 2.0 | | | | | COCO-QA | | | | |
|---|---|---|---|---|---|---|---|---|---|---|
| Method | B@4 | C | M | R | QB@1 | B@4 | C | M | R | QB@1 |
| I2Q | 9.02 | 63.21 | 13.89 | 35.33 | 26.32 | 13.53 | 95.90 | 12.61 | 36.23 | 31.37 |
| IT2Q | 18.41 | 134.88 | 19.90 | 45.71 | 40.27 | 17.80 | 128.64 | 16.17 | 43.22 | 38.88 |
| IMVQG | 19.72 | 149.28 | 20.43 | 46.76 | 40.40 | 18.43 | 127.18 | 17.21 | 44.07 | 39.22 |
| Dual | 19.90 | 151.60 | 20.60 | 47.00 | 41.90 | 18.80 | 131.10 | 17.73 | 44.19 | 39.92 |
| Radial | 20.70 | 161.90 | 21.40 | 48.10 | 43.50 | 19.24 | 139.55 | 18.19 | 44.21 | 40.98 |
| Ours | **22.43** | **180.55** | **22.57** | **49.36** | **45.61** | **20.84** | **166.78** | **19.81** | **46.80** | **43.57** |

We claim that the comparison with other baselines under this setting is quite fair since we just use data pre-processing to generate the visual hints. Furthermore, the generated visual hints are quite noisy since we align it by GloVe cosine similarity instead of the exact matching.

**Pre-processing** We apply pre-trained ResNet101 and Masked-RCNN to extract features. And we truncate questions longer than 20 words and build vocabulary on words with at least 3 occurrences. Since the test split is not open for the public, we divide the train set to 90% train split and 10% validation split. Please refer to Appendix C for more detailed pre-processing and implement settings.

## 4.2 BASELINE METHODS AND EVALUATION METRICS

**Baseline Methods** We compare against the following baselines in our experiments: i) I2Q* (means image to question), ii) IT2Q (means images with answer type hints to question), iii) IMVQG* (Krishna et al., 2019), iv) Dual* (Li et al., 2018), v) Radial* (Xu et al., 2020). The Detailed description of these baselines is provided in Appendix B. Experiments on baselines followed by * are conducted using the codes originally released by authors.

**Evaluation Metrics** Following previous works, we adopt the standard linguistic measures including BLEU (Papineni et al., 2002), CIDEr (Vedantam et al., 2015), METEOR (Banerjee & Lavie, 2005), ROUGE-L (Lin, 2004) and Q-BLEU (Nema & Khapra, 2018). For visual hints prediction, we adopt F1 metric. And for answer and object position prediction tasks, we use accuracy and mean-IOU (Intersection Over Union) metrics. These scores are calculated by official evaluation scripts.

### 4.2.1 RESULT ANALYSIS AND HUMAN EVALUATION

Table 1 shows the automatic evaluation results comparing our proposed method against other state-of-art baselines. Note that we presents the linguistic evaluation metrics and the visual-hints, answer and object position prediction tasks' performance are omitted due to limited space, please refer to more details in Appendix D. We can find that our method consistently outperforms previous methods by a significant margin on both datasets. It highlights that our proposed method which asking questions with double hints achieves state-of-art performance, thus address the two issues we highlighted in Sec. 1.

Furthermore, we conduct a human evaluation study to assess the quality of the questions generated by our method, the ground truth (GT) and the baselines: Radial and Dual in terms of syntax, semantics, and relevance metrics. The results are shown in Table 3. Details are shown in Appendix F. We can see that our model outperforms all the strong baseline methods by a large margin on all metrics. And we observe slightly inferior results on the semantics metric and a margin on the relevance metric. So we conduct error analysis which is shown in the following section.

## 4.3 ABLATION STUDY

In this section, we conduct the ablation studies to demonstrate the effect of double hints, double-hints guided graph, visual attention, and multi-task auto-encoder on the VQA 2.0. Concretely, we will remove one or more components at one time to generate the ablation model as follows: (1) w/o. visual hints (abbr: -visual hints): we remove the visual-hints prediction module. (2) w/o. double hints (abbr:

-visual/answer hints + answer type): we remove both the visual and answer hints but use the answer-type to further assess the effect of double hints. (3) w/o. double-hints guided graph (abbr: -gnn): we remove the implicit graph construction and gcn encoding modules. (4) w/o. visual attention (abbr: -visual attention): we remove the non-visual-hint mask during attention when decoding. (5) w/o. double-hints guided graph and visual attention (abbr: gnn, visual attn): We combine the ablation (3) and (4) together to further assess the effect of them. (6) w/o. position and answer auto-encoder (abbr: -pos/ans predict): we set $\beta$ and $\gamma$ to zero for reducing multi-task auto-encoder to visual hints only based auto-encoder.

The ablation study results on VQA2.0 val set's results are shown in Table 2. Firstly, the full model outperforms all the ablation models. It confirms that the visual hints, multi-task auto-encoder, and the graph are useful components.

Specifically, by turning off the visual hints, the model's performance drops nearly 2.8% (METEOR). It confirms that the visual hints are indeed helpful for high-quality question generation. And when we further discard the answer hints, the performance continues to drop rapidly by 18.2% (BLEU@4), which further demonstrates that the answer hints are a more helpful signal compared to the answer type. These results indicate the importance of double hints.

In addition, by turning off the double-hints guided graph, we observe that the performance drops nearly 1.2% (METEOR). It demonstrates that it is beneficial to exploit the hidden relations by learning a dynamic graph. Furthermore, when we discard the visual attention during decoding, the performance drops by 1.3% (METEOR). This result illustrates that if we force the model to focus on only the predicted visual-hint object regions, it can generate higher quality questions, which further confirms the impact of visual hints. Additionally, if we turn off both the graph and the visual attention, the performance continues to drop, which further confirms that both these two components play an indispensable role in the framework. Finally, we observe that if we discard the multi-task auto-encoder, the performance drops slightly. It shows that by predicting the object position and target answer, the model can learn more robust visual hint features.

Table 2: Ablation Results on VQA2.0.

| Method | B@4 | M | QB@1 |
|---|---|---|---|
| Full Model(Double hints) | **22.43** | **22.57** | **45.61** |
| - visual hints | 21.67 | 21.93 | 44.52 |
| - visual/answer hints + answer type | 18.53 | 20.06 | 40.32 |
| - gnn | 22.17 | 22.31 | 45.34 |
| - visual attn | 21.98 | 22.26 | 45.11 |
| - gnn,visual attn | 21.92 | 22.21 | 45.04 |
| - pos/ans predict | 22.34 | 22.51 | 45.57 |

Table 3: Human-study Results on VQA2.0.

| Method | Syntax | Semantics | Relevance |
|---|---|---|---|
| Radial | 4.25 (0.36) | 4.04 (0.45) | 3.09 (0.24) |
| Dual | 4.53 (0.40) | 4.38 (0.58) | 3.24 (0.26) |
| Ours | **4.73 (0.35)** | 4.61 (0.40) | 3.58 (0.39) |
| GT | 4.52 (0.34) | **4.63 (0.41)** | **4.14 (0.50)** |

## 4.4 CASE STUDY AND ERROR ANALYSIS

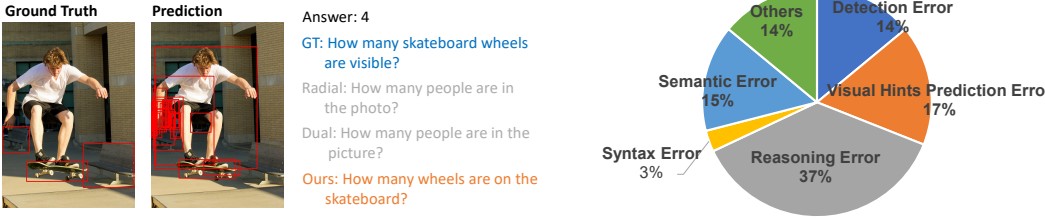

Figure 2: Case study (left) and error analysis (right) results.

In this section, we will further show one example to illustrate the quality of our method compared with the same baselines. What's more, we also dive into the failures of our model by analyzing the error cases. The results are shown in Figure 2, for more concrete results please refer to Appendix G.

As we can see, by learning the object graph and generating with the guidance of double hints, we can find that our model indeed generates more answer-aware, region-referential, and high-quality questions.

And for error cases, it is very difficult to classify a certain example as a certain error cause, we can only roughly divide it into five categories: 1) visual hint prediction error: It means the error made by visual hints prediction and further mislead the generation. 2) detection error: It means our model detects the objects mistakenly. 3) reasoning error: It means our model makes the wrong relation reasoning. 4) syntactic error: It means there are grammar errors. 5) semantic error: It means the question has drawbacks in the semantic view. Further detailed examples and analysis are shown in Appendix H.

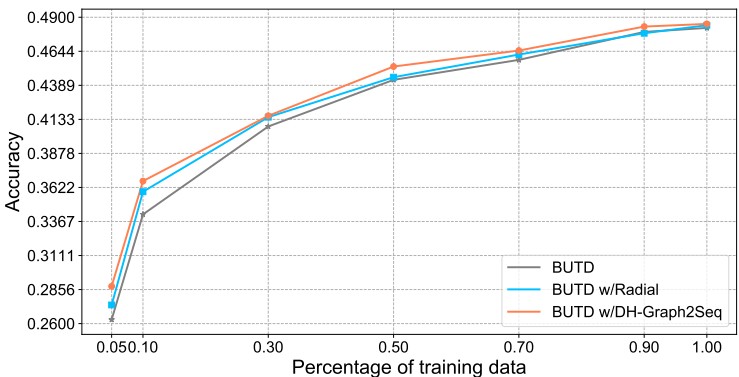

Figure 3: Performance of VQA model in under different proportions of supervised data.

### 4.5 DATA AUGMENTATION

One of the most important applications of VQG is providing more training data for VQA. In this section, we use the proposed model to generate more questions for training VQA methods. Firstly, we adopt the classic Bottom-Up Top-Down (abbr: BUTD) (Anderson et al., 2018) as the VQA benchmark on VQA 2.0 dataset and use top-1 accuracy as the evaluation metric. In order to examine the effect of QG-driven data augmentation on the VQA, we compare the performance of the BUTD baseline with two data augmentation variants, namely, BUTD w/Radial and BUTD w/DH-Graph2Seq. Specifically, we split the train-set (the same as the VQG dataset) to $x\% \in \{0.05, 0.1, 0.3, 0.5, 0.7, 0.9, 1\}$. The BUTD baseline is trained only on the $x$ part, while the other two variants are trained on the combination of the golden $x$ part and the questions generated by VQG models. What's more, to train the VQG model, we further split $x$ part to $80\%/20\%$(train/dev). The results are shown in 3.

From the results, we observe that both VQG models consistently help improve VQA performance. Notably, our DH-Graph2Seq model outperforms the Radial baseline consistently. In particular, when $x$ is relatively smaller, which means that under a low-source condition, data-augmentation is more effective.

## 5 CONCLUSION

In this paper, we propose a new setting with double hints for the visual question generation task. Under the new setting, we further design a novel DH-Graph2Seq model which can explore the correlations among the objects in the image generate the questions with answer-awareness and region-reference. Our extensive experiments on VQA2.0 and COCO-QA datasets demonstrate that our proposed model can significantly outperform existing state-of-the-art by a large margin.

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

## A    THE DETAILS OF GRAPH EMBEDDING

We apply multi-layer GCN with residual connection (He et al., 2016) to effectively learn the node embedding from the constructed object graph. The basic layer is shown as follows:

$$\mathbf{X}^{out} = \frac{\sigma(\mathbf{D}^{-1/2}\hat{\mathbf{A}}\mathbf{D}^{-1/2}\mathbf{X}^{in}\mathbf{W}) + \mathbf{X}^{in}}{\sqrt{2}} \tag{10}$$

where $\mathbf{X}^{in}$ is the node feature ($\mathbf{X}$ in Eq.7), $\hat{\mathbf{A}}$ is the adjacency matrix ($\mathbf{A} + \mathbf{I}$), $\mathbf{D}$ is the degree matrix of $\hat{\mathbf{A}}$, $\mathbf{W}$ is the trainable weights and $\sigma$ is the ReLU activation function.

## B    THE DETAILS OF BASELINE MODELS

**I2Q**    It means generating the questions without any hints. We adopt the classic image caption show attend and tell method (Xu et al., 2015).

**IT2Q**    It means generating questions with answer types. We modify the show attend and tell (Xu et al., 2015) method to take input from the joint embedding of the image and answer type. And since there are no answer type annotations in the original datasets, we adopt the same answer-type information as IMVQG which is a baseline we will discuss later.

**IMVQG (Krishna et al., 2019)**    This is a baseline that maximizes the mutual information among the generated questions, the input images, and the expected answers. They use the answer category (type) as a hint. For VQA 2.0 dataset, since they just annotate 80% of the original dataset, so we annotate the rest of them as "other". And for the COCO-QA dataset, we find that there are only 430 answers, so we annotate their type attribute by ourselves just like they do in VQA 2.0.

**Dual (Li et al., 2018)**    This is another competitive baseline that views the VQG task as the dual task of VQA based on MUTAN architecture. They train the VQG task along with VQA to enhance both VQG and VQA's performance.

**Radial (Xu et al., 2020)**    This is the latest strong baseline for VQG. They use answers to build an answer-radial object graph and learn the graph embedding. Then they use the graph2seq paradigm to generate the questions.

## C    IMPLEMENTATION DETAILS

### C.1    DATA PROCESSING

We use a pre-trained ResNet-101 model to extract visual image features. And for each image, we use a Masked-RCNN detector with ResNeXt-101 backbone to detect 100 object regions (selected by confidence score) and feature extraction (fc6). The ResNet-101 is from torchvision and the Masked-RCNN is from Detectron2.

### C.2    HYPER-PARAMETER SETTINGS

**Overall setting**    The image feature dimension is 2048 and the object feature dimension is 1024. The words' dimension is 512 and their weights are randomly initialized. The hidden size in Eq. 3 is 128 to save CUDA memory. All the hidden size is 1024 if not otherwise specified. As for the visual hints prediction, the $\eta$ is 4 and $\lambda$ is 2. The overall loss function's $\alpha$ is 0.5, $\beta$ is 0.01 and $\gamma$ is 0.01 for most cases. But for ablation model '-vh', $\alpha$ is 0, $\beta$ is 0.01 and $\gamma$ is 0.01. For ablation model '-position', $\alpha$ is 0.5, $\beta$ is 0 and $\gamma$ is 0.01. And for ablation model '-answer', $\alpha$ is 0.5, $\beta$ is 0.01 and $\gamma$ is 0. We adopt adam (Kingma & Ba, 2014) optimizer with 0.0002 learning rate. The batch size is 120.

**Graph embedding module setting**    The multi-head cosine similarity metric's $k$ is 3. The layer of GCN is 2. The sparsing hyper-parameter $\epsilon$ is 0.75.

Table 4: Results on VQA2.0 val set. All accuracies are in %.

| Method | BLEU@4 | CIDEr | METEOR | ROUGE | Q-BLEU@1 | F1 | Accuracy | mIOU |
|--------|--------|-------|--------|-------|----------|-----|----------|------|
| I2Q | 9.02 | 63.21 | 13.89 | 35.33 | 26.32 | - | - | - |
| IT2Q | 18.41 | 134.88 | 19.90 | 45.71 | 40.27 | - | - | - |
| IMVQG | 19.72 | 149.28 | 20.43 | 46.76 | 40.40 | - | - | - |
| Dual | 19.90 | 151.60 | 20.60 | 47.00 | 41.90 | - | - | - |
| Radial | 20.70 | 161.90 | 21.40 | 48.10 | 43.50 | - | - | - |
| Ours | **22.43** | **180.55** | **22.57** | **49.36** | **45.61** | 63.24 | 82.4% | 0.752 |

## D    THE DETAILS OF RESULTS

See Figure 4 and 5 for full results.

## E    ANALYSIS OF HYPERPARAMETERS

### E.1    THE $\epsilon$ ANALYSIS

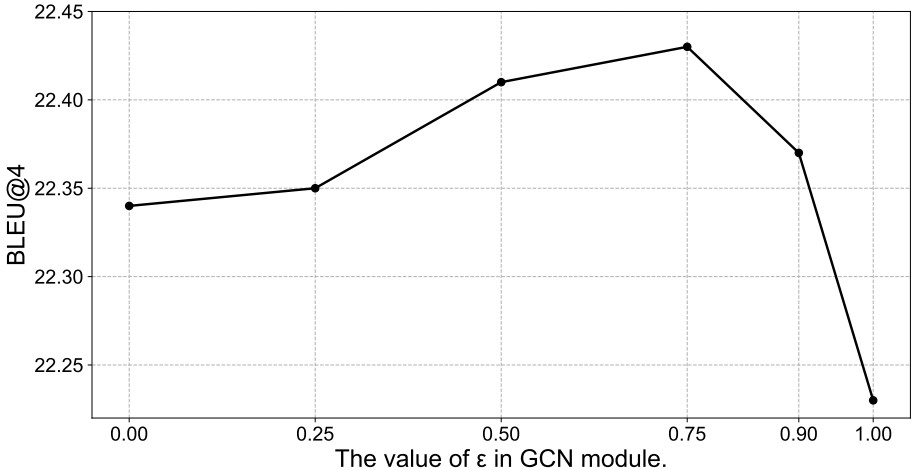

Figure 4: The effect of different graph sparsity.

To study the effect of the $\epsilon$ in sec. 3.3, we conduct experiments on VQA 2.0 with the $\epsilon$ varying in $[0, 1]$. The results are shown in Figure 4. We find that the model achieves the best performance when $\epsilon$ ranging in $[0.5, 0.8]$. It drops rapidly when $\epsilon$ is close to 1 which means the graph is too sparse.

### E.2    THE $\mu$ ANALYSIS

To study the effect of the $\mu$ in sec. 4.1, we conduct experiments on VQA 2.0 with the $\mu$ varying in $\{3, 5.7, 7\}$. When $\mu$ is larger, the visual hint will be more accurate, but the reserved questions will be much less. So in order to compare the effect fairly, we use the same test set and evaluate both the DH-Graph2Seq model and DH-Graph2Seq w/o visual hint (abbr: DH-Graph2Seq w/o vh). The results are shown in Figure 5. We can find that the DH-Graph2Seq consistently outperforms the DH-Graph2Seq w/o vh. When $\mu$ is smaller, although the visual hints are more accurate, the processed dataset has less questions which lead to performance drop. And when $\mu$ is larger, although we can reserve more questions, the visual hints' quality is lower. So we think a reasonable choice of $\mu$ is near 5.7.

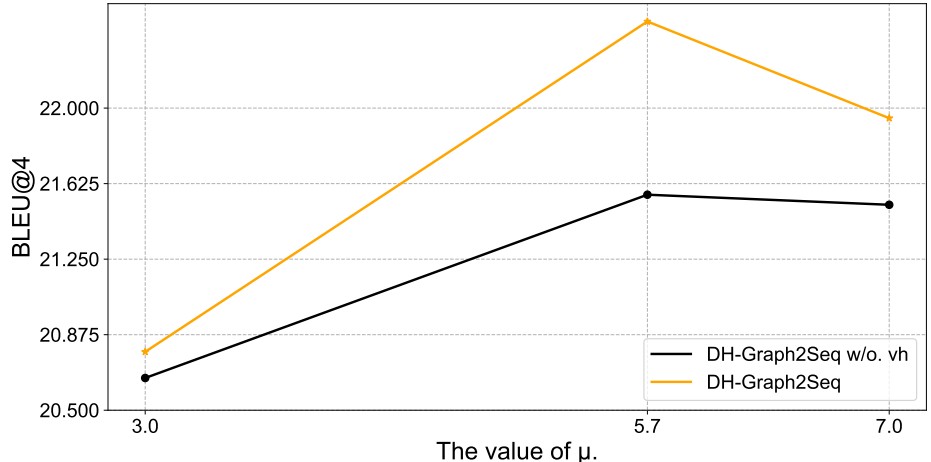

Figure 5: The effect of $\mu$.

Table 5: Results on COCO-QA val set. All accuracies are in %.

| Method | BLEU@4 | CIDEr | METEOR | ROUGE | Q-BLEU@1 | F1 | Accuracy | mIOU |
|--------|--------|-------|--------|-------|----------|-----|----------|------|
| I2Q | 13.53 | 95.90 | 12.61 | 36.23 | 31.37 | - | - | - |
| IT2Q | 17.80 | 128.64 | 16.17 | 43.22 | 38.88 | - | - | - |
| IMVQG | 18.43 | 127.18 | 17.21 | 44.07 | 39.22 | - | - | - |
| Dual | 18.80 | 131.10 | 17.73 | 44.19 | 39.92 | - | - | - |
| Radial | 19.24 | 139.55 | 18.19 | 44.21 | 40.98 | - | - | - |
| Ours | **20.84** | **166.78** | **19.81** | **46.80** | **43.57** | 60.81 | 85.2% | 0.713 |

# F    THE DETAILS OF HUMAN EVALUATION

We conduct a small-scale human evaluation on the VQA2.0 val set. Concretely, we randomly select 50 examples for each system: 1) the ground-truth results (denoted as GT), 2) our results (denoted as Ours), 3) the 'Radial' baseline's results (denoted as Radial), 4) the 'dual' baseline's results (denoted as Dual).

We ask 5 human evaluators to give feedback on the quality of questions randomly selected in the results of 4 systems. In each example, given a triple containing a raw image, a target answer, and an anonymized system's output, they are asked to rate the quality of the output by answering the three questions: a) is the question syntactically correct? b) is the question semantically correct? c) is the question relevant to the image and the answer pair? For each question, the rating scale is from 1 to 5. The standard is 1. Pool (not acceptable), 2. Marginal, 3. Acceptable, 4. Good, 5. Excellent. We develop software to automatically collect the evaluation results shown in Figure 6. The software will feed the examples and average the scores.

# G    THE DETAILS OF CASE STUDY

See Figure 7 for qualitative examples. We can find that our model can generate more complete and vivid questions compared with baseline methods. We think that with the help of double hints, our graph2seq model can find the proper image regions and exploit the rich structure relations better.

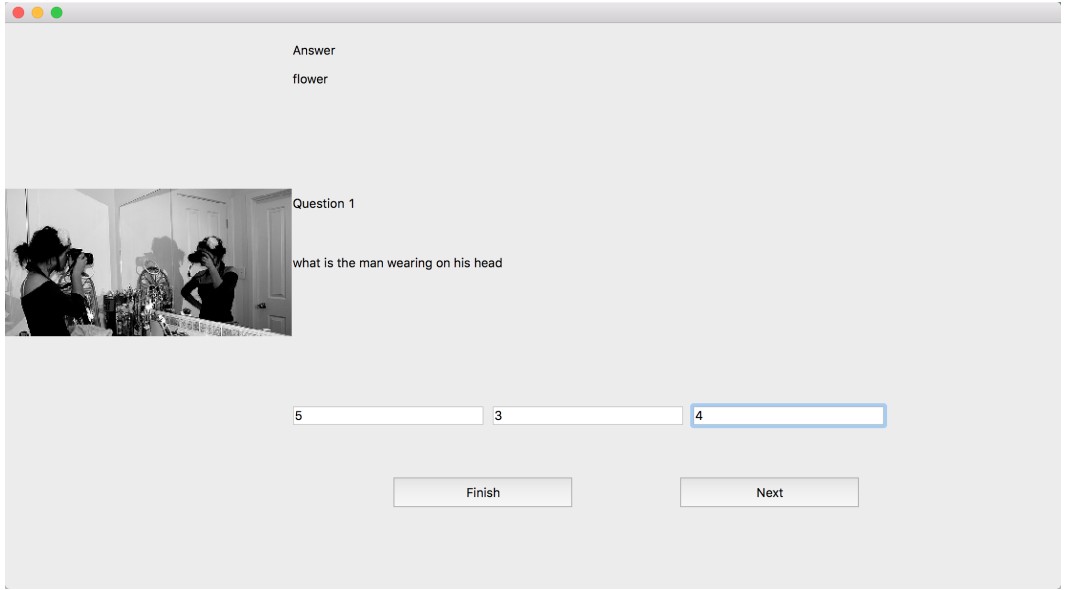

Figure 6: The screenshot of human evaluation software.

## H    THE DETAILS OF ERROR ANALYSIS

See Figure 8 for error cases of our results. We present one example of each error reason.

a) It means our model recognizes the objects incorrectly. In the picture, the 'teddy bear' is next to the refrigerator but our model recognizes it as 'microwave'.

b) It means our model predicts the visual hints incorrectly, which largely misleads the generation. Indeed, the answer '3' refers to the number of humans, but the model picks the horses out and overlooks the men.

c) It means our model infers the relations among the objects incorrectly. In the image, the batter wears the helmet but the expected answer is 'no'.

d) It means our model generates questions which are semantically incorrect. In the image, there are at least two cars, but in the generated question, the car is in the singular form.

e) It means our model generates questions which are syntactically incorrect. The 'How many of' is incorrect.

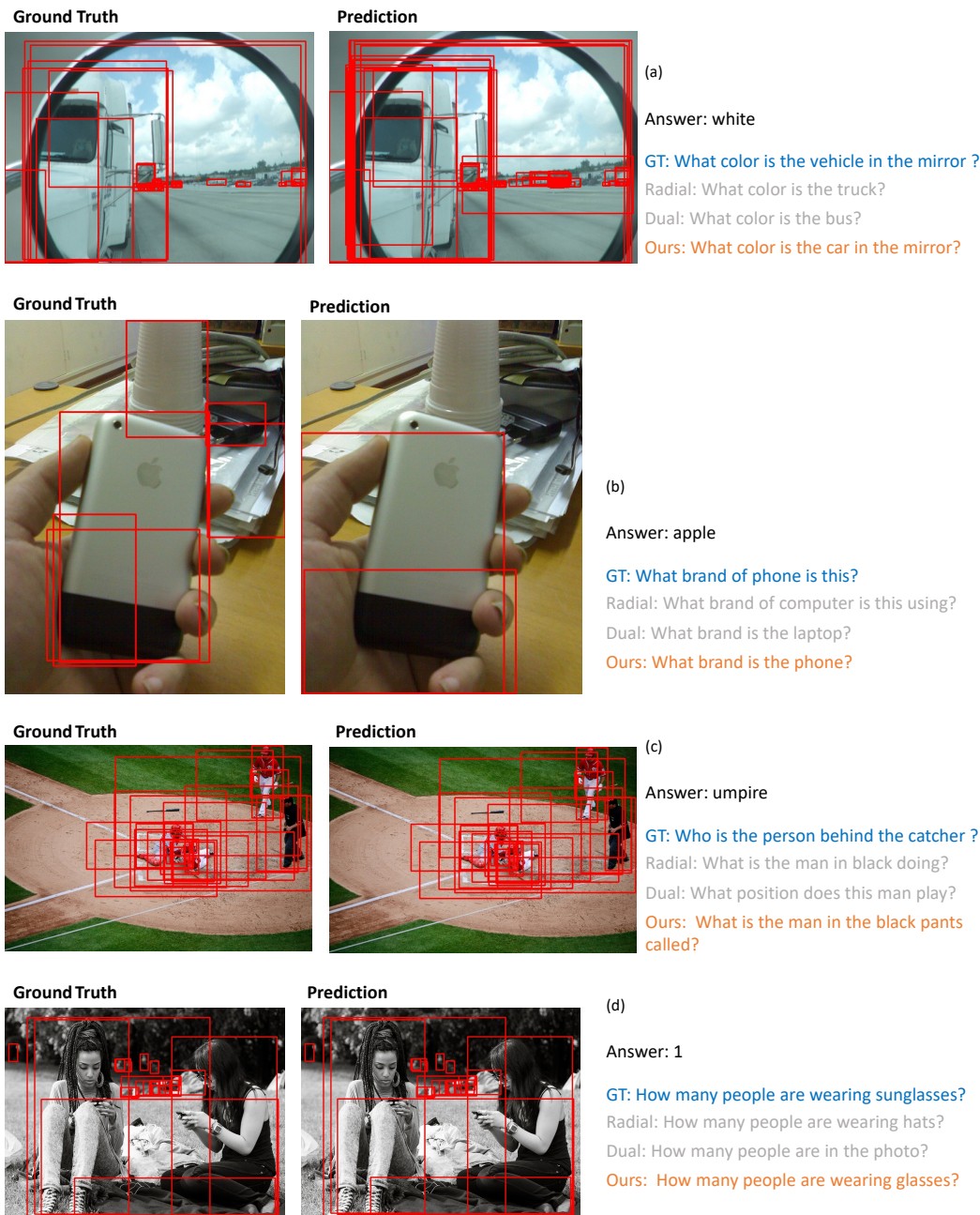

Figure 7: The details of case study examples. The red rectangles mean the visual-hint regions.

**a)**

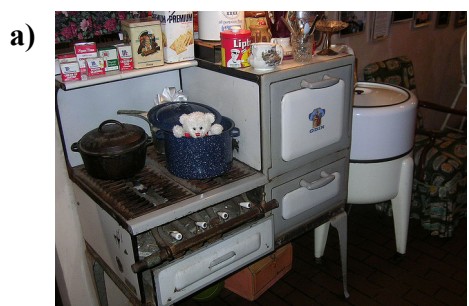

**Detection Error**

A: teddy bear
Q: What is sitting on the shelf next to the microwave?

**b)**

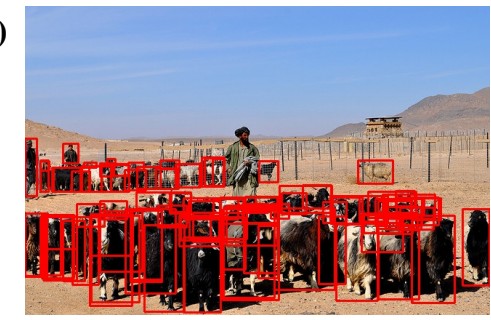

**Visual Hints Prediction Error**

A: 3
Q: How many horses are not white?

**c)**

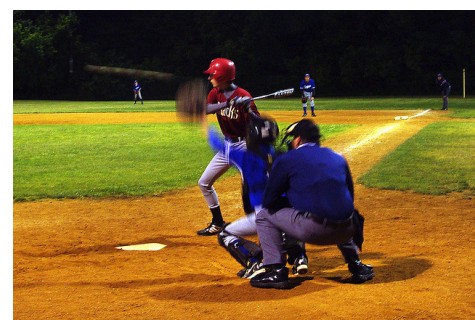

**Reasoning Error**

A: No
Q: Is the batter wearing a helmet?

**d)**

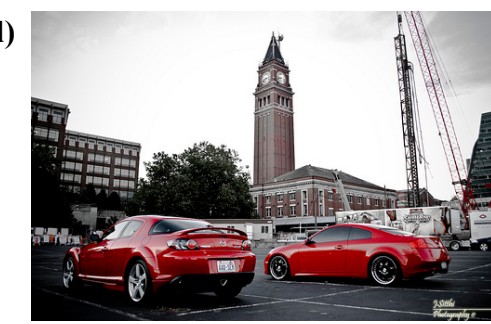

**Semantic Error**

A: Red
Q: What color is the car?

**e)**

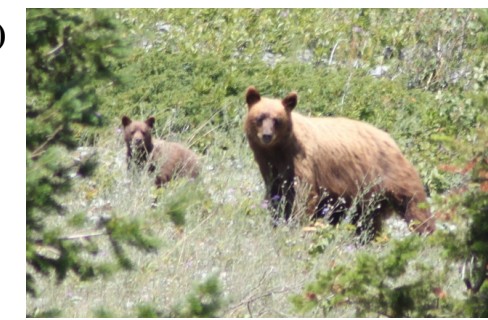

**Syntactic Error**

A: 2
Q: How many of bears are there?

Figure 8: The details of error examples.

