# OpenReview forum: "Ask Question with Double Hints:  Visual Question Generation with Answer-awareness and Region-reference"
_ICLR.cc/2021/Conference — Reject_

### Official Review · AnonReviewer2 · 2020-10-26
**An OK submission, but the motivation is questionable. And some details need to be clarified.**

**Rating:** 6
**Confidence:** 5

**Review:**

Overview:
This submission focuses on the problem of visual question generation and proposes to use two hints: answer and visual regions. With the help of these two hints, a graph representation is built and then a GCN-based graph2sequence generator is applied to generate questions based on given inputs. Experiments on two datasets show its efficiency on question generation quality.

Strengths:
+ The description of the method (model) is easy to follow.
+ Improvements of the question generation quality (according to the BLEU and Cider score) is big.
+ Good ablation studies and human study.

Weakness:
- The motivation of VQG: I understand VQG is a recently raised task in vision-and-language research, but I can not get any ideas that why this is an important task, at least from this paper. According to my understanding, one reason that we need VQG is that we can use it as a data-augmentation tool to train a better VQA system (which is a more important task). Another reason is that it may lead to a better  (goal-oriented) visual dialogue system, such as GuessWhat?!.  This paper mentioned these two directions very briefly but failed to explain it well and there are no experiments presented in this paper to show whether their question generation model can benefit these two areas. I don't think a better BLEU score can show its potential since it only means this model can generate similar questions as the training data. It is worth to check whether these augmented questions can boost VQA performance.

- The motivation of two hints: It looks like many previous works only use answer types as the hint. I think this is more reasonable since it gives more freedom to the model to generate questions. And I don't think it is an issue that 'one answer/image can be potentially mapped to many different questions'. Instead, I think this provides a diversity of generated questions and further improves the VQA generalisation ability, from a data augmentation view. In this submission, authors choose to prove the answer directly as a hint. There might be two issues: at first, it seems not fair since it requires more annotation. Secondly, how can you get the answer during the inference? Say if one downloads an image from the Internet  (not from a well-annotated VQA dataset) and there are three bears in the image, it might be easy if the answer is 'bear' since you can use a detector, but how can one know the answer is 'three' if we want to generate a question about 'how many?'. In this case, the answer type is more practical.

According to the visual hints, if I am right, from section 4.1, they are mined from the questions and answers directly, for both training and testing. This suggests you (authors) already got some info from the target questions (even in the testing!). I can't say it is a fair setting, even it happens in pre-processing step. Please fix me if I am wrong.

- The 3.2.1 is basically an attention mechanism?

- What is the accuracy of the tasks listed in 3.2.2?

- In section 3.3, the \epsilon seems quite important. It decides the sparsity of the graph. Any ablation studies?

- In section 4.1, another \epsilon, which decides the quality of the visual hints. Any ablation studies?

--------------------------- After Rebuttal --------------------------------

The authors did a good job in the rebuttal. Most of my concerns have been addressed so I am happy to raise my score to 6.

---

> ### Author Response · Authors · 2020-11-18
> **Author response to Review #2**
>
> We would like to thank the reviewer for your careful reading and providing a lot of valuable comments! Below we address the concerns mentioned in the review:
>
> Q1: The motivation of VQG...I don't think a better BLEU score can show its potential since it only means this model can generate similar questions as the training data. It is worth to check whether these augmented questions can boost VQA performance.
>
> We are sorry that we did not make the motivation of VQG very clear in our initial submission. Yes, indeed we fully agree that these automatic metrics like BLEU is not enough to measure the quality of generated questions. Therefore, we also provide human study. And furthermore, following your suggestion, we also use VQG as data augmentation for its dual task VQA is a necessary step to measure the quality of the outputs of VQG. We have further conducted a new set of experiments to test whether two VQG methods can help improve the VQA task.
>
> We adopt the classic Bottom-Up Top-Down(abbr: BUTD) as the VQA benchmark on VQA 2.0 dataset and use top-1 accuracy as the evaluation metric. For more details, please refer to the figure 3 in the revision (sec 4.5) for the full results and settings.
>
> Results:
>
> | model name                       | 0.05   | 0.1     | 0.3     |  0.5     |
>
> | BUTD                                   | 0.263 | 0.342 | 0.408 |  0.443 |
>
> | BUTD w/Radial                  | 0.274 | 0.359 | 0.415 |  0.445 |
>
> | BUTD w/DH-Graph2Seq  | 0.288 | 0.367 | 0.416 | 0.453  |
>
> As shown in the table, we observe that both VQG models consistently help improve VQA performance. Notably, our DH-Graph2Seq model outperforms the Radial baseline consistently. In particular, when x is relatively smaller (the number of training samples is small) , which means that under a low-source condition, data-augmentation is more effective.
>
> Finally, we would like to point out that QG models are also particularly useful for the few-shot learning or zero-shot learning, as shown in some very recent papers [1][2]. However, this is out of scope this paper tried to study and we will leave it as one of the future works.
>
> [1] Liangming Pan, Wenhu Chen, Wenhan Xiong, Min-Yen Kan, William Yang Wang, “Unsupervised Multi-hop Question Answering by Question Generation”, https://arxiv.org/abs/2010.12623.
>
> [2] Siamak Shakeri, Cicero Nogueira dos Santos, Henry Zhu, Patrick Ng, Feng Nan, Zhiguo Wang, Ramesh Nallapati, Bing Xiang, “End-to-End Synthetic Data Generation for Domain Adaptation of Question Answering Systems”, https://arxiv.org/abs/2010.06028.
>
>
> Q2a: The motivation of two hints
>
> In general, the history of VQG in the literature has evolved over the last few years after the VQA task was introduced and can generally be divided into three categories in order: 1) no hints; 2) answer-type hint; 3) answer hint.
>
> For the VQG with no hints, although more diverse questions can be generated, there are no corresponding answers either. As argued in [Ranjay et. al.], a successful QG system should be “goal-driven” in order to provide useful additional QA pairs to further improve existing VQA generalization ability from a data augmentation view.
>
> To address this issue, researchers later proposed to use answer-type hints for VQG. In all VQA datasets, it is common to have several questions with overlapped question types (what, how, how many and so on) for one image (one-to-many mapping issue). If only answer-type is used, it will either generate not answerable questions or specific questions, which either cannot be used for training VQA or reduce diversity of questions for improving generalizability of VQA.
>
> To further address this issue, researchers subsequently proposed to use answer hints directly for VQG. Since even though two questions could be the same question types for the same image, the answer typically can greatly reduce the ambiguities. Therefore, it can train a better question generator for more diverse questions instead of specific questions. However, there are still many cases where answer hints are not enough to reduce the ambiguities for generating a question (please see our responses of Q3 to Review #3).
>
> Our double hints approach is a novel learning paradigm for an important vision+language task - visual question generation (VQG), to effectively solve two severe issues with existing approaches: 1) fail to address the existing one-to-many mapping issues (solved by the proposed Double Hints - textual answer and visual regions of interests); 2) fail to consider the fine-grained relations among region features (solved by learned dynamic implicit object graph with proposed DH-Graph2Seq model). To the best of our knowledge, using Double Hints for VQG has not been exploited before in the previous works. We believe that the combination of Double Hints, Construction of sparse implicit graphs, Casting VQG as graph-to-sequence learning problem is novel and researchers who are working on Vision + NLP will find our approach beneficial.

---

> ### Author Response · Authors · 2020-11-18
> **Author response to Review #2 (continued)**
>
> Q2b: There might be two issues: at first, it seems not fair since it requires more annotation. Secondly, how can you get the answer during the inference?
>
> There are some debating about the use of answer type and answer hints for VQG in [[Ranjay et. al., Yikang et. al.]. We would like to highlight that since the main purpose of VQG is to generate more data samples for existing VQA datasets there are no requirements for additional annotations. It is just about what and how to leverage these answer information. For our visual hints, we generate the (noisy) annotations by using automatic NLP and CV tools (see sec. 4.1) so it requires no additional human annotations. Therefore, it is a fair comparison between our approach and existing methods.
>
> [1] Ranjay Krishna, Michael Bernstein, and Li Fei-Fei. Information maximizing visual question generation. In Proceedings of the IEEE Conference on Computer Vision and Pattern Recognition, pp.2008–2018, 2019.
>
> [2] Yikang Li, Nan Duan, Bolei Zhou, Xiao Chu, Wanli Ouyang, Xiaogang Wang, and Ming Zhou. Visual question generation as dual task of visual question answering. In Proceedings of the IEEE Conference on Computer Vision and Pattern Recognition, pp. 6116–6124, 2018
>
> Q3: According to the visual hints, if I am right, from section 4.1, they are mined from the questions and answers directly, for both training and testing. This suggests you (authors) already got some info from the target questions (even in the testing!). I can't say it is a fair setting, even it happens in pre-processing step. Please fix me if I am wrong.
>
> This is a misunderstanding. The visual hints are only provided in the training stage. They are predicted by the model during the inference (see sec. 3.2). Therefore, the comparison is fair since the visual-hints are mined from <Q, A> pairs by automatic tools and there are no additional human annotations.
>
>
> Q4: The 3.2.1 is basically an attention mechanism?
>
> Technically, it is a type of attention or cross-attention. This module is designed to exploit the hidden relations among the answer clues, image features, and object features.
>
>
> Q5: What is the accuracy of the tasks listed in 3.2.2?
>
> We added the answer accuracy and mean-IOU results in Appendix D due to limited pages.
>
> Q6: In section 3.3, the \epsilon seems quite important. It decides the sparsity of the graph. Any ablation studies?
>
> Yes, the \epsilon in sec. 3.3 decides the sparsity of the graph. We conduct the ablation study and the results are shown in the Appendix E.1. We find that the model achieves the best performance when \epsilon ranging in [0.5, 0.8]. The performance drops when \epsilon is either close to 0 or 1 which means the graph is either fully dense or too sparse.
>
> The results:
>
> | \epsilon |    BLEU@4  |
>
> |     0       |        22.26    |
>
> |    0.25   |       22.35     |
>
> |   0.5      |        22. 41   |
>
> |   0.75    |        22.43    |
>
> |   0.9      |        22.37    |
>
> |    1        |        22. 23   |
>
>
> Q7: In section 4.1, another \epsilon, which decides the quality of the visual hints. Any ablation studies?
>
> We are sorry that we made a mistake in the original submission. It is not cosine similarity but the l2 distance. To further study the quality of the visual hints, we conduct an ablation study in Appendix E.2.
> When \mu is larger, the visual hint will be more accurate, but the number of the candidate questions will be much less. So in order to compare the effect fairly, we use the same test set and evaluate both the DH-Graph2Seq model and DH-Graph2Seq w/o visual hint (abbr: DH-Graph2Seq w/o vh). We can find that the DH-Graph2Seq consistently outperforms the DH-Graph2Seq w/o vh. When \mu is smaller, although the visual hints are more accurate, the processed dataset has less questions which lead to performance drop. And when \mu is larger, although we can have more candidate questions, the visual hints' quality is lower. In our experiment, a reasonable choice of \mu is near 5.7, which is a hyperparameter to be tuned.
>
> | \epsilon |    BLEU@4-novh  |  BLEU@4 |
>
> |     3       |        20.66              |    20.79    |
>
> |   5.7      |        21.57              |    22.43    |
>
> |     7       |        21.52              |    21.95    |

---

### Official Review · AnonReviewer3 · 2020-10-27
**Limited novelty and the poor writing makes this paper a rejected one**

**Rating:** 5
**Confidence:** 5

**Review:**

In this paper, the authors aim at generating the right questions based on the textual answers and corresponding visual regions of interest (ROIs). The core innovation of the proposed method is leveraging the region information to supervise the question generation, which helps to mitigate the ambiguity of the answers. Correspondingly, a simple method is designed to generate the noisy annotations of the ROIs using the pretrained Masked-RCNN and the questions.

This core part is mainly divided into two components: 1) aligning the object features with the answer word embeddings using attention mechanism; 2) constructing the object graphs and getting the GCN-refined object features; 3) refining the question embeddings by attending the refined object features and image ones. Experimental results show that the proposed method outperforms the existing approaches significantly. In addition, ablation studies prove the effectiveness of the proposed components.

Although the visual regions of interest can help to guide the question generation, the one-to-many mapping issues still exist when generating the visual regions of interest (no additional information is given), which may lead to the same problem in the question generation stage. Nevertheless, the proposed approach is still valuable. Although it helps little in mitigating the one-to-many issues, it helps to learn a better question generation network, as the additional region information can mitigate the issue during the training stage, which will help to avoid learning a generator that prefers to give general questions. Therefore, I think authors should make that clear in the paper that it helps in learning a better question generator instead of generating the specific question because the ROIs are also selected based on the answers only.

Although great improvements are achieved, the proposed method is not novel. The main novelty is leveraging the object features in generating questions. However, neither the feature alignment nor the GCN part is new. In addition, the graph construction part does not make sense to me, and no ablation study show that pruning the non-hints objects can help to improve the performance.

In addition, the confusing equations and poor writing of the paper cannot make this paper an unaccepted one:
* In Eq. (2), \beta X^{a} seems like a matrix, which cannot be concatenated with a vector.
* In Eq. (3), \beta_{j} should be \beta_{i, j}.
* In Eq. (3), what I_{j} means? Why the image feature has a subscript?
* In Eq. (3), what does the operation “*” means? The product signs are different in (2) (3) and (4)
* In Sec. 3.4, it lacks detailed explanations of how the model will attend the image and graph to get a better representation of the question embedding between the two LSTMs, which is one of the core components of the proposed methods.
* Typos like “size information” should be “side information” in the top paragraph on Page 2, “question tokens” should be “word tokens” in the bottom paragraph on Page 3.

---

> ### Author Response · Authors · 2020-11-18
> **Author response to Review #3**
>
> We are grateful to the reviewer for a nice summary, and for the kind recognition of our key contributions. We thank you for providing a lot of critical comments as well! Below we address the concerns mentioned in the review:
>
> Q1: Although great improvements are achieved, the proposed method is not novel. The main novelty is leveraging the object features in generating questions. However, neither the feature alignment nor the GCN part is new.
>
> Our most important contribution is to present a novel learning paradigm for an important vision+language task - visual question generation (VQG), to effectively solve two severe issues with existing approaches: 1) fail to address the existing one-to-many mapping issues (solved by the proposed Double Hints - textual answer and visual regions of interests); 2) fail to consider the fine-grained relations among region features (solved by learned dynamic implicit object graph with proposed DH-Graph2Seq model). To the best of our knowledge, using Double Hints for VQG has not been exploited before in the previous works. We believe that the combination of Double Hints, Construction of sparse implicit graphs, Casting VQG as graph-to-sequence learning problem is novel and researchers who are working on Vision + NLP will find our approach beneficial. In addition, to learn these visual hints, we develop a multi-task auto-encoder to learn the visual hints and the unique attributes automatically without introducing any additional human annotations. We are the first to explore this kind of study in order for automatic visual hint learning without expensive and time consuming human annotations.
>
>
> Q2: In addition, the graph construction part does not make sense to me, and no ablation study show that pruning the non-hints objects can help to improve the performance.
>
> Our graph construction part is designed to capture the rich interactions between visual and answer hints and the image as well as the sophisticated relationships among the visual objects in an image, which is generally ignored by the previous works. Our ablation study to show the effectiveness of each component - GNN and visual attention (pruning the non-hints objects) below.
>
> | Method  	           	                | BLEU@4   |  METEOR  | QBLEU@1   |
>
> |  Full Model          		              | 22.43         |  22.57         | 45.61           |
>
> |- visual hints         		             | 21.67         |  21.93         | 44.52           |
>
> |- double hints + answer type  | 18.53         |  20.06         | 40.32           |
>
> |- gnn		                                    |  22.17        |  22.31         | 45.34           |
>
> |- visual attn	                            | 21.98         |  22.26         | 45.11           |
>
> Q3: Although the visual regions of interest can help to guide the question generation, the one-to-many mapping issues still exist when generating the visual regions of interest (no additional information is given), which may lead to the same problem in the question generation stage. Nevertheless, the proposed approach is still valuable. Although it helps little in mitigating the one-to-many issues, it helps to learn a better question generation network, as the additional region information can mitigate the issue during the training stage, which will help to avoid learning a generator that prefers to give general questions. Therefore, I think authors should make that clear in the paper that it helps in learning a better question generator instead of generating the specific question because the ROIs are also selected based on the answers only.
>
> We appreciate the reviewer's very thoughtful comments here. However, we respectfully disagree. Our double hints Graph2Seq model can help learn a better question generator for generating more diverse questions because the ROIs (visual hints) can provide additional information beyond answer hints (although they are generated using answer hints).
>
> In the motivating example in image 1, we chose an example that the answer hint seems almost to coincide with visual hint, which may give you this expression that visual hints does not help much in mitigating the one-to-many mapping issues. Let’s take another image example where some people are skating on a mountain. One person is facing the camera while another person is away from the camera. But both of them wear black clothes. The answer here is black and we need to ask a right question given this answer. In this case, we can either question “what color is the man facing the camera wearing?” or question “what color is the man away from the camera wearing?” In this example, the visual hints are crucial for reducing the ambiguities in order to ask a right question. In other words, it helps mitigate the one-to-many mapping issues with the visual hint beside the answer hint. We will add more examples in the appendix in order to make double hints motivation more clear.

---

> ### Author Response · Authors · 2020-11-18
> **Author response to Review #3 (continued)**
>
> Q4: In addition, the confusing equations and poor writing of the paper...
>
> We really appreciate your careful reading and point out these typos and unclear parts. We have polished the paper in the revision.

---

### Official Review · AnonReviewer1 · 2020-10-30
**Good results on existing benchmarks; Concerns around novelty and key take-away messages.**

**Rating:** 6
**Confidence:** 5

**Review:**

Summary:
The paper proposes a model for the task of Visual Question Generation (VQG) which uses the answer as well as object regions to generate the question. The paper models interactions between various visual entities and the answer tokens using a graph and then use it to generate the question. The proposed approach outperforms existing methods on the VQA and COCO-QA task.

Strengths:
- The paper emphasizes on using both image-regions for the VQG task. While some of the proposed techniques borrows from existing works, they showed how to combine it to improve over exisitng methods.

- On the VQA and COCO benchmarks, the models outperforms existing methods. On the small scale human study, the proposed approach was rated higher than existing approaches.

Weaknesses:
- Overall, the novely of the paper is low. The paper is a collection of already popular ideas (use of attented region features for V+L tasks (Shah et al.), use of position embeddings to model spatial relations (ViLBERT, VL-BERT etc). Because, the paper is written as a collection of ideas, the take-away message isn't clear.

- The ablation tables has very subtle performance changes across design choices which are hard to understand. The ablation table suggests that all choices in isolation leads to a drop when compared to the full model. But what happens when you combine these choices one by one. More importantly, what are the most critical components of the proposed method?

- Some of the statements made in the paper are not backed by citations or lack explanation. For instance, the authors claim that the "amount of objects which are visual hints are much smaller than ones not" and "when humans ask questions from the image, we will infer whether the object is important clues by looking around in the image".

- The paper was hard to follow and it seems like there are a lot of moving parts. In general, this makes reproducing the paper and adopting ideas from the paper difficult.

---

> ### Author Response · Authors · 2020-11-18
> **Author response to Review #1**
>
> We want to thank the reviewer for your careful reading and providing a lot of critical comments! Below we address the concerns mentioned in the review:
>
> Q1: Overall, the novelty of the paper is low. The paper is a collection of already popular ideas (use of attented region features for V+L tasks (Shah et al.), use of position embeddings to model spatial relations (ViLBERT, VL-BERT etc). Because the paper is written as a collection of ideas, the take-away message isn't clear.
>
> Our most important contribution is to present a novel learning paradigm for an important vision+language task - visual question generation (VQG), to effectively solve two severe issues with existing approaches: 1) fail to address the existing one-to-many mapping issues (solved by the proposed Double Hints - textual answer and visual regions of interests); 2) fail to consider the fine-grained relations among region features (solved by learned dynamic implicit object graph with proposed DH-Graph2Seq model). To the best of our knowledge, using Double Hints for VQG has not been exploited before in the previous works. We believe that the combination of Double Hints, Construction of sparse implicit graphs, Casting VQG as graph-to-sequence learning problem is novel and researchers who are working on Vision + NLP will find our approach beneficial. In addition, to learn these visual hints, we develop a multi-task auto-encoder to learn the visual hints and the unique attributes automatically without introducing any additional human annotations. We are the first to explore this kind of study in order for automatic visual hint learning without expensive and time consuming human annotations.
>
> We would like to clarify the difference between our work and several mentioned works as well. The use of attended region features and position embeddings are not new and as we highlighted for most important contributions these techniques are not our main contributions but just some basic elements of our overall proposed learning framework.
>
>
> Q2: The ablation tables has very subtle performance changes across design choices which are hard to understand... More importantly, what are the most critical components of the proposed method?
>
> This is a great suggestion! Following your suggestions, we have conducted a set of new experiments and completely reconstruct the ablation study to show the effectiveness of each component more clearly in the revision (sec. 4.3).
>
> | Method  	           	                    | BLEU@4   |  METEOR  | QBLEU@1   |
>
> |  Full Model          		                 | 22.43         |  22.57         | 45.61           |
>
> |- visual hints         		                | 21.67         |  21.93         | 44.52           |
>
> |- double hints + answer type     | 18.53         |  20.06         | 40.32           |
>
> |- gnn		                                       | 22.17        |  22.31         | 45.34           |
>
> |- visual attn	                               | 21.98         |  22.26         | 45.11           |
>
> |- gnn,visual attn	                       | 21.92         |  22.21         | 45.04           |
>
> |- pos/ans predict                         | 22.34         |  22.51          | 45.57           |
>
> As shown in the above Table (on VQA2.0), the full model outperforms all the ablation models. We can easily observe that the double hints, visual hints, gnn (for encoder), and visual attention (for decoder) are all important components. As expected, double hints play a key role in final performance as we see that once we replace double hints with only answer type all performance metrics drop significantly. Interestingly, if we discard the multi-task auto-encoder, the performance drops slightly. But we think that by predicting the object position and target answer, the model can learn more robust visual hint features.
>
>
> Q3: Some of the statements made in the paper are not backed by citations or lack explanation.
>
> Thanks for your suggestions. We have clarified and properly cited the corresponding references in the revision.
>
>
> Q4: The paper was hard to follow and it seems like there are a lot of moving parts. In general, this makes reproducing the paper and adopting ideas from the paper difficult.
>
> We will release our codes upon the paper acceptance. What’s more, the specific hyper-parameter settings are provided in Appendix C.2.

---

### Official Review · AnonReviewer4 · 2020-11-02
**New model on Visual Question Generation**

**Rating:** 6
**Confidence:** 3

**Review:**

The paper introduces a new model on the task of Visual Question Generation. The model uses cross-modal alignment between the object features, position features and answer hints to find the right subset of relevant visual hints to be used to generate the relevant question. The model also ensures that the latent space features capture the answer and position information by predicting them back from it. Informed by the visual hints, the object and image features are passed to a GCN network that is used to get the final hidden state which is passed to an attention and language lstm similar to BUTD model to generate the final question.

The cycle consistency used for answer and position features seems to provide grounded question generation for the answer and image. The model is tested on VQA2.0 and COCOQA and achieves better performance compared to the baseline model on automated metrics as well as human evaluation.

Overall, the paper is strong and provides a solid foundation for the intuition and framework behind the model supported with detailed ablation analysis and case studies. What I find missing, is the actual test of how good these questions actually are by using them to train on the actual VQA task. Understanding that and how it performs as extra data on VQA 2.0 would give us a better understanding on how good the generation actually is where it matters.

Some other questions:
- Is there any specific reason why BERT wasn’t used instead of GloVe for word embeddings?
- Please add citation for VQA 2.0

Overall, I would like to recommend the paper for acceptance but it is hard to understand the actual value of this work without downstream application on the task of VQA 2.0.

Edit after rebuttal: I have read the author response and I thank the authors for their valuable insights and answers to my questions. It is exciting that this does help in improving the performance on downstream VQA2.0 task though I would have expected the results to be conducted on one of the recent state-of-the-art models instead of very old BUTD model where achieving performance gains is trivial. I would like to keep my rating as it is.

---

> ### Author Response · Authors · 2020-11-18
> **Author response to Review #4**
>
> First of all, we are very grateful to the reviewer for the accurate summary, and for the kind recognition of our key contributions. We also want to thank the reviewer for your thorough reading and valuable comments!
>
> Q1: What I find missing, is the actual test of how good these questions actually are by using them to train on the actual VQA task.
>
>
> This is a great suggestion! We fully agree that using VQG as data augmentation for its dual task VQA is a necessary step to measure the quality of the outputs of VQG. Following your suggestions, we have further conducted a new set of experiments to test whether two VQG methods can help improve the VQA task.
>
> Firstly, we adopt the classic Bottom-Up Top-Down(abbr: BUTD) as the VQA benchmark on VQA 2.0 dataset and use top-1 accuracy as the evaluation metric. In order to examine the effect of QG-driven data augmentation on the VQA, we compare the performance of the BUTD baseline with two data augmentation variants, namely, BUTD w/Radial and BUTD w/DH-Graph2Seq. Specifically, we split the train-set (the same as the VQG dataset) to x \in {0.05, 0.1, 0.3, 0.5, 0.7, 0.9, 1}. The BUTD baseline is trained only on the x part, while the other two variants are trained on the combination of the golden x part and the questions generated by two VQG models. What's more, to train the VQG model, we further split x part to 80%/20%(train/dev). Here we list the results when x lies in {0.05, 0.1, 0.3, 0.5}, please refer to the figure 3 in the revision (sec 4.5) for the full results.
>
> Results:
>
> | model name                                  | 0.05   | 0.1     | 0.3     |  0.5     |
>
> | BUTD                                               | 0.263 | 0.342 | 0.408 |  0.443 |
>
> | BUTD w/Radial                              | 0.274 | 0.359 | 0.415 |  0.445 |
>
> | BUTD w/DH-Graph2Seq              | 0.288 | 0.367 | 0.416 | 0.453  |
>
> As shown in the table, we observe that both VQG models consistently help improve VQA performance. Notably, our DH-Graph2Seq model outperforms the Radial baseline consistently. In particular, when x is relatively smaller (the number of training samples is small) , which means that under a low-source condition, data-augmentation is more effective.
>
> Finally, we would like to point out that QG models are also particularly useful for the few-shot learning or zero-shot learning, as shown in some very recent papers [1][2]. We would like to exploit these directions in the near future.
>
> [1] Liangming Pan, Wenhu Chen, Wenhan Xiong, Min-Yen Kan, William Yang Wang, “Unsupervised Multi-hop Question Answering by Question Generation”, https://arxiv.org/abs/2010.12623.
>
> [2] Siamak Shakeri, Cicero Nogueira dos Santos, Henry Zhu, Patrick Ng, Feng Nan, Zhiguo Wang, Ramesh Nallapati, Bing Xiang, “End-to-End Synthetic Data Generation for Domain Adaptation of Question Answering Systems”, https://arxiv.org/abs/2010.06028.
>
>
> Q2: Is there any specific reason why BERT wasn’t used instead of GloVe for word embeddings?
>
> We used GloVe word embedding instead of pretrained-BERT mainly for a fair comparison with our baselines (where all of them used GloVe). Conceptually, we did not see any issue to apply pretrained-BERT word embeddings in our model. In order to test the performance of pretrained-BERT word embeddings, we conduct an experiment using BERT as the initial word embedding and show the results as follows:
>
> BERT experiment in VQA 2.0 dataset’s result:
>
>  | model_name | BLEU@4   | CIDEr    | METEOR | ROUGE | Q-BLEU@1 |
>
>  | Ours-GloVe   |  22.43        | 180.55   | 22.57       | 49.36     |  45.61           |
>
>  | Ours-BERT   |  22.46        | 180.61   | 22.56       | 49.37     |  45.61           |
>
> Interestingly, we did not find significant performance increase with BERT but instead two variants’ performance is pretty close. We think that it may be because the answer sequence is too short (i.e. no more than three words) and in many cases there is only one word (e.g. yes/no). So the effect of BERT is limited.
>
>
> Q3:Please add citation for VQA 2.0
>
> We have added the VQA 2.0 and some other missed citations in the revision.

---

### Author Response · Authors · 2020-11-24
**Follow-up**

Dear Reviewers,

Has our response addressed all your concerns? Let us know if you have more questions. Thank you!

---

### Decision · Program_Chairs · 2021-01-07
**Final Decision**

**Decision:**

Reject

**Comment:**


The paper proposes to generate human-like question for a image by using additional information (hints) such as the textual answer and visual regions of interest (ROIs).  The visual regions are used to guide the question generation so that the model can generate relevant and informative question.  The question generation problem is formulated as a graph-to-sequence problem, starting from an object graph, and using GCNs with attention to align text and visual regions and to generate an appropriate question.

Review Summary: The submission initially mixed reviews (scores from 3 to 6).  While reviewers find the problem interesting and work to be mostly solid, reviewers felt that the novelty of the work was limited (R1, R3) and that some of the presentation was unclear (R1, R3) with some missing details (R2).  R2 was not sure if the VQG was a useful task, and R4 felt that the initial submission was missing a key experiment on whether the generated questions can be useful as data augmentation for VQA. The reviewers were impressed by the extra experiments performed by the authors in the rebuttal and the revised draft, and indicated that most of their concerns were addressed, In particular,  the generated questions were shown to be useful as data augmentation.  Many reviewers increased their scores, ending with 3 scores of 6 (R1,R2,R4) and 1 score of 5 (R3).

Pros:
- The use of generated questions as data augmentation for VQA is an interesting direction
- Strong empirical results with thorough experiments (and user study)

Cons:
- The technical novelty of the work is still rather limited (R1)
- The paper was difficult to follow (R1,R3) with a lot of moving parts, making it potentially difficult reproduce (R1) - The authors indicated that they will open-source the code.
- The grammar and wording of the writing is poor even after revision and should be improved

Example of poor writing (a full proofreading pass is recommended):
- Section 2.1: "Mora et al. (2016) firstly makes an attempt to adapt" => "Mora et al. (2016) adapted",
"abstract and general results" => "imprecise and generic questions"
- Section 3.2: "The most important point of our first issue located in how to effectively find..." (it's unclear what this means)
- Section 4.1: "standfordcoreNLP" => "Stanford CoreNLP"
- Section 4.5: "shown in 3" => "shown in Figure 3"

Recommendation:
The AC agrees that the work is addresses an interesting area of generating questions as data augmentation for VQA.  Despite the improved reviewer scores, the AC agrees with the initial assessment that the work has limited technical novelty.  The AC also found the writing of the paper to be poor and difficult to follow at places even after revision.  Due the limited novelty, the issues with exposition, and the many changes to paper, the AC believe that the work would benefit from another round of review and should not be accepted at ICLR in its current form.  Given the positive response, the authors are encourage to improve their work and writing and resubmit to an appropriate venue (the AC believes the work would be more appreciated in a vision or language venue).